# Comparison of Kit-Based Metabolomics with Other Methodologies in a Large Cohort, towards Establishing Reference Values

**DOI:** 10.3390/metabo11100652

**Published:** 2021-09-24

**Authors:** Daisuke Saigusa, Eiji Hishinuma, Naomi Matsukawa, Masatomo Takahashi, Jin Inoue, Shu Tadaka, Ikuko N. Motoike, Atsushi Hozawa, Yoshihiro Izumi, Takeshi Bamba, Kengo Kinoshita, Kim Ekroos, Seizo Koshiba, Masayuki Yamamoto

**Affiliations:** 1Department of Integrative Genomics, Tohoku University Tohoku Medical Megabank Organization, 2-1 Seiryo-machi, Aoba-ku, Sendai 980-8573, Japan; ehishi@ingem.oas.tohoku.ac.jp (E.H.); matsukawa@megabank.tohoku.ac.jp (N.M.); jinoue@megabank.tohoku.ac.jp (J.I.); tadaka@megabank.tohoku.ac.jp (S.T.); motoike@megabank.tohoku.ac.jp (I.N.M.); kengo@ecei.tohoku.ac.jp (K.K.); koshiba@megabank.tohoku.ac.jp (S.K.); masiyamamoto@med.tohoku.ac.jp (M.Y.); 2Medical Biochemistry, Tohoku University Graduate School of Medicine, 2-1 Seiryo-machi, Aoba-ku, Sendai 980-8575, Japan; 3Advanced Research Center for Innovations in Next-Generation Medicine, Tohoku University, 2-1 Seiryo-machi, Aoba-ku, Sendai 980-8573, Japan; 4Division of Metabolomics, Medical Institute of Bioregulation, Kyushu University, 3-1-1 Maidashi, Higashi-ku, Fukuoka 812-8582, Japan; m-takahashi@bioreg.kyushu-u.ac.jp (M.T.); izumi@bioreg.kyushu-u.ac.jp (Y.I.); bamba@bioreg.kyushu-u.ac.jp (T.B.); 5Graduate School of Information Sciences, Tohoku University, 6-3-09, Aramaki Aza-Aoba, Aoba-ku, Sendai 980-8579, Japan; 6Department of Preventive Medicine and Epidemiology, Tohoku University Tohoku Medical Megabank Organization, 2-1 Seiryo-machi, Aoba-ku, Sendai 980-8573, Japan; hozawa@megabank.tohoku.ac.jp; 7Department of Systems Life Sciences, Graduate School of Systems Life Sciences, Kyushu University, 3-1-1 Maidashi, Higashi-ku, Fukuoka 812-8582, Japan; 8Lipidomics Consulting Ltd., 02230 Espoo, Finland; kim@lipidomicsconsulting.com

**Keywords:** metabolic profiling, widely-targeted metabolomics, lipidomics, mass spectrometry, UHPLC-MS/MS, UHPLC-FT/MS, SFC-MS/MS, large-scale cohort

## Abstract

Metabolic profiling is an omics approach that can be used to observe phenotypic changes, making it particularly attractive for biomarker discovery. Although several candidate metabolites biomarkers for disease expression have been identified in recent clinical studies, the reference values of healthy subjects have not been established. In particular, the accuracy of concentrations measured by mass spectrometry (MS) is unclear. Therefore, comprehensive metabolic profiling in large-scale cohorts by MS to create a database with reference ranges is essential for evaluating the quality of the discovered biomarkers. In this study, we tested 8700 plasma samples by commercial kit-based metabolomics and separated them into two groups of 6159 and 2541 analyses based on the different ultra-high-performance tandem mass spectrometry (UHPLC-MS/MS) systems. We evaluated the quality of the quantified values of the detected metabolites from the reference materials in the group of 2541 compared with the quantified values from other platforms, such as nuclear magnetic resonance (NMR), supercritical fluid chromatography tandem mass spectrometry (SFC-MS/MS) and UHPLC-Fourier transform mass spectrometry (FTMS). The values of the amino acids were highly correlated with the NMR results, and lipid species such as phosphatidylcholines and ceramides showed good correlation, while the values of triglycerides and cholesterol esters correlated less to the lipidomics analyses performed using SFC-MS/MS and UHPLC-FTMS. The evaluation of the quantified values by MS-based techniques is essential for metabolic profiling in a large-scale cohort.

## 1. Introduction

A metabolome is a group of small molecules that are endogenously produced as part of the end of the central dogma and have biological functions; metabolomics can be used together with other omics techniques [1]. Metabolic changes are directly associated with phenotypic changes and are affected by genomic factors, environmental factors (such as lifestyle, food intake, and/or the gut microbiome), and disease expression and progression [2,3]. Therefore, metabolic phenotyping has been widely used in biomarker discovery studies to identify disease-specific predictive molecules in biological specimens using several analytical platforms [4,5,6].

Nuclear magnetic resonance (NMR) has conventionally been used for metabolic profiling because of its high-quality quantification [7,8]. In contrast, mass spectrometry (MS)-based metabolic profiling allows for the simultaneous and sensitive detection of metabolites, and gas chromatography MS has been traditionally utilized for this purpose [9,10]. However, it is impossible to extract both hydrophilic and hydrophobic molecules together, such as amino acids and lipid species, within a single sample preparation procedure using single solvent systems. Global metabolomics has been established to detect thousands of features as a comprehensive metabolic profiling technique using liquid chromatography MS (LC/MS) for biomarker discovery, although the disadvantages of lower reproducibility, lower accuracy of quantification and increased effort to annotate the structures of the molecules while considering biological functions remain [11,12].

Recently, kit-based metabolomics (Kit-Met) using LC/MS, which is in strict accordance with a standard operating procedure involving detailed documentation for sample preparation, instrument setup, system suitability testing, and data analysis, was established, which enabled us to obtain quantified values of several hundred metabolites and to compare the quantified values with interlaboratory studies [13,14]. Metabolic profiling can be performed using a kit consisting of a consumable 96-well preparation plate that includes several internal standards (ISs) and the optimal ultrahigh-performance liquid chromatography triple quadrupole tandem mass spectrometry (UHPLC-MS/MS) methods with two separate methodologies. Based on technological developments, more than 600 metabolites can be detected by means of the current Kit-Met version. In fact, representative metabolites, including amino acids, amino acid-related metabolites, bile acids, biogenic amines, cresol, fatty acids, hormones, indole derivatives, nucleobases, and vitamins, can be detected in UHPLC-MS/MS mode using an analytical column, and most lipid species (such as acylcarnitines, ceramides (Cers), cholesterol esters (CEs), diacylglycerols (DGs), dihydroceramides, and glycerophospholipids (including lysophosphatidylcholines (LPCs) and phosphatidylcholines (PCs), glycosylceramides, sphingolipids, sugars, and triacylglycerols (TG)), can be detected in flow injection analysis (FIA)-MS/MS mode from biological samples. These kits have recently been used in clinical studies for the discovery of biomarkers for diseases in a large number of patients worldwide, such as patients with mild cognitive impairment (MCI) [15], Alzheimer’s disease [16], Parkinson’s disease [17], depression [18], autism [19], chronic obstructive pulmonary disease [20], cardiovascular disease [21], diabetes mellitus [22], chronic kidney disease [23], glaucoma [24], lung cancer [25], hepatocellular carcinoma [26], gastric cancer [27], head and neck cancer [28], and breast cancer [29]. However, only disease-based samples have been used in most metabolic profiling clinical studies, which did not include optimal subjects adjusted for age, sex, or body mass index (BMI) because of the limitations of obtaining samples from healthy subjects in the hospital. Therefore, it is necessary to compare the metabolic profiles of healthy controls selected with similar phenotypes to improve the sensitivity and specificity of the biomarkers for clinical examination applications.

National biobank projects have incorporated metabolic profiling in large-scale analyses in not only clinical cohorts but also prospective cohorts [16,30,31], and this information has been stored for epidemiologic analysis with genetic variation. However, method standardization, which is important for cross-study and cross-cohort comparisons in metabolic profiling, remains a challenge for LC/MS-based widely targeted metabolomics techniques [32]. Notably, interlaboratory studies have demonstrated that most lipid species in plasma vary greatly due to the use of different platforms and laboratories during the lipid consortium project [33]. Hydrophilic metabolite outliers in cell lysates can be observed depending on the analytical method [34]. Therefore, establishment of a reference database for metabolic profiling, which will allow comparison with other methods and laboratories, has been required for a long time.

The Tohoku Medical Megabank (TMM) Project was established as one of the largest cohort projects in the Tohoku area of Japan, which was affected by earthquakes and a tsunami disaster on 11 March 2011, and several biospecimens obtained from more than 150,000 participants are stored at the biobank [35]. One of the main goals of this cohort was to identify predictive biomarkers of disease expression for future precision medicine by using metabolic profiling techniques combined with other datasets, such as genomics, phenotypes, and/or habitudes [36,37]. During this large-scale cohort metabolic profiling project, forty-five metabolites from more than 30,000 plasma samples were analyzed by NMR, 110 metabolites from approximately 2000 plasma samples and 421 metabolites from approximately 2300 plasma samples were analyzed by Kit-Met [6], and their quantified values were included in the commercial database “Japanese Multi Omics Reference Panel, jMorp” [38]. The database has expanded both the number of samples and the number of quantified metabolites, and the quantified values from cohort participants could potentially be utilized as references for biomarker discovery studies.

In this study, 8700 plasma samples with reference materials were analyzed by Kit-Met. We first subjected the 6159 plasma samples to Kit-Met 1 and then subjected the 2541 plasma samples to Kit-Met 2. Both Kit-Met 1 and Kit-Met 2 analyses were performed with the same kit (MxP^®^ Quant 500 kit) using different UHPLC-MS/MS systems: Xevo^®^ TQ-S and Xevo^®^ TQ-XS MS/MS systems for Kit-Met 1 and Kit-Met 2, respectively. The instrument used for Kit-Met 2 is slightly more sensitive than that used for Kit-Met 1 and has the potential to expand the number of quantified metabolites. The values of the detected metabolites in the reference materials were first evaluated via Kit-Met 1 and Kit-Met 2. We then evaluated the quality of the quantified values of the metabolites in 2541 plasma samples detected by Kit-Met 2 compared with the quantified values from other platforms, such as NMR, supercritical fluid chromatography MS/MS (SFC-MS/MS) and UHPLC-Fourier transform MS (UHPLC-FTMS) systems. We demonstrated the utilization of Kit-Met profiling in a large-scale cohort with interplate normalization by principal component analysis (PCA).

## 2. Results

A summary of the present study is shown in Figure 1. A total of 8700 plasma samples from participants in the TMM Community-based Cohort Study were selected for metabolic profiling using Kit-Mets by UHPLC-MS/MS. The demographic characteristics of the participants, which were separated into two groups of 6159 and 2541 plasma samples based on the difference in UHPLC-MS/MS setups, are described in Table 1. In this study, metabolic profiling of the 6159 and 2541 plasma samples was performed by Kit-Met 1 and Kit-Met 2, respectively (Table 2). Kit-Met 2 was used as a system with higher MS sensitivity than Kit-Met 1. NIST^®^ SRM^®^ 1950 plasma samples and four global quality control (gQC) plasma samples (pooled normal human plasma, Na EDTA) along with 77 cohort plasma samples were analyzed on each 96-well plate using the MxP^®^ Quant 500 kit.

### 2.1. Evaluation of System Difference in the Large-Scale Analysis

#### 2.1.1. Comparison of the Variation Performed Using Kit-Met 1 and Kit-Met 2

We first examined 6159 cohort plasma samples with the NIST and gQC plasma samples and observed the coefficient of variation (CV, %) of each quantified metabolite to evaluate the variation in quantified values by Kit-Met analyses. During data analysis, we selected only quantification values of higher quality, which were defined by the Kit-Met criteria, as highlighted in green and blue in Appendix A, and the detected metabolites were analyzed separately in UHPLC-MS/MS mode and FIA-MS/MS mode.

In UHPLC-MS/MS mode, most metabolites, including amino acids, amino acid-related metabolites, bile acids, biogenic amines, cresol, fatty acids, hormones, indole derivatives, nucleobases, and vitamins, were widely detected around several ranges of the CV in the NIST plasma samples by Kit-Met 1, and a total of 36 metabolites were quantified inside 30% CV (Figure 2a, upper left panel, black bars). In addition, the peak frequency of numerous metabolites, including lipid species (such as acylcarnitines, Cers, CEs, DGs, dihydroceramides, and glycerophospholipids (including LPCs and PCs, glycosylceramides, sphingolipids, sugars and TG)), was detected at approximately 30% of the CV in the NIST samples in FIA-MS/MS mode by Kit-Met 1, indicating that large variations could be observed by Kit-Met 1, and a total of 314 metabolites were quantified inside 30% CV (Figure 2b, upper right panel, black bars).

We therefore examined 2541 plasma samples by Kit-Met 2 to improve the data quality of the quantified values of the metabolites. Indeed, most metabolites could be observed within 10% of the CV in the NIST plasma samples in UHPLC-MS/MS mode by Kit-Met 2, and a total of 75 metabolites, including amino acids and their related metabolites, were quantified inside 30% CV (Figure 2a, left panel, white bars). The peak frequency of the number of lipid species shifted from 30% to 10% in FIA-MS/MS mode by Kit-Met 2, and a total of 376 metabolites, including LPCs, PCs, sphingolipids, and TGs, were quantified inside 30% CV (Figure 2b, right panel, white bars). The CV differences between Kit-Met 1 and Kit-Met 2 were similarly observed with the quantified values of the metabolites in the gQC plasma samples (Appendix A). The list of mean values of metabolites sorted by CV values in NIST and gQC plasma is shown in Appendix A.

These differences in variation demonstrated that the metabolite values quantified by Kit-Met 2 were more reliable than those quantified by Kit-Met 1 in our cohort study. However, there was a threefold higher number of plates run on Kit-Met 1, which tends to increase variance in a large-scale cohort study. Although the median intensities of the boxplot for detected metabolites in NIST plasma could not observe significant differences among 80 plates evaluated by Kit-Met 1 and 33 plates by evaluated Kit-Met 2 (Appendix Aa–d), the single plots of select molecules (such as glutamine and PCaa 34:2) had the largest variation among 80 plates evaluated by Kit-Met 1 analyses, even after normalization by gQCs (Figure 2c,d).

#### 2.1.2. Correlation Analysis of the Quantified Values of Metabolites in the NIST Plasma Samples Detected by NMR and Kit-Met 1 and Kit-Met 2 in UHPLC-MS/MS Mode

We then examined the NIST plasma samples by NMR analysis by following our previously described protocol [39]. Fifty-two metabolites detected from the NIST plasma samples by NMR are summarized in Appendix A. Thirty-one metabolites detected by Kit-Mets were correspondingly quantified with NIST plasma samples by NMR. The plot of the ratio of the concentration between the mean values of metabolites detected by Kit-Met 1 or Kit-Met 2 and their mean values detected by NMR with CV (%) demonstrated that approximately half of the metabolites were highly correlated with the quantified values determined by NMR within the lower CV (dotted circle colored red, Figure 3a,b, upper panels). However, cysteine (Cys), arginine (Arg), lysine (Lys), lactic acid (Lac), ornithine (Orn), p-cresol sulfate (p-Cresol-SO_4_), and hypoxanthine were detected at twofold higher levels than by NMR, whereas hippuric acid (HipAcid), indoxyl sulfate (Ind-SO_4_), and trimethylamine oxide (TMAO) levels were lower than those detected by NMR (Figure 3a,b, upper panels). Interestingly, the average concentrations of 32 metabolites, which were selected by a CV < 30% in 80 and 33 NIST plasma samples using Kit-Met 1 and Kit-Met 2, respectively, were highly correlated (*r* > 0.997, Spearman correlation analysis) (Figure 3a, bottom left panel). Therefore, the absolute quantified values of those metabolites detected by Kit-Mets should be evaluated by other technology to create a database as a reference.

#### 2.1.3. Correlation Analysis of the Quantified Values of Metabolites in the NIST Plasma Samples Detected by Kit-Met 1 and Kit-Met 2 in FIA-MS/MS Mode

We next evaluated the quantified values of lipid species by Kit-Mets. The 314 lipid species were detected within a CV < 30% in 132 NIST plasma samples by Kit-Met 1, while 376 lipid species were stably detected in 33 NIST plasma samples by Kit-Met 2 (Figure 2b). The difference was simply derived from the interplate variation and sensitivity of the MS system; quantified values of 273 lipid species were codetected within a CV < 30% from the NIST plasma samples, and the lipid class-dependent distribution (such as PCs and CEs) could be observed on the plot of correlation between Kit-Met 1 and Kit-Met 2 by FIA-MS/MS (Figure 3d, bottom right panel).

We therefore evaluated the values of lipid species separating lipid classes quantified by Kit-Met 2, which were detected with higher quality compared to other platforms used for lipidomic profiling. In addition, the consensus values of the lipid species in the NIST plasma samples were validated by an interlaboratory study [33]. We thus examined the NIST plasma samples by using SFC-MS/MS and UHPLC-FTMS to evaluate the quality of the lipid species values quantified by FIA-MS/MS mode using Kit-Met 2, and the values from the three platforms were compared with the consensus values previously described [33].

### 2.2. Evaluation of the Quantified Values of the Lipid Species in the NIST Plasma Samples

#### 2.2.1. Correlation Analysis of the Quantified Values of the Lipid Species in the NIST Plasma Samples by SFC-MS/MS and FIA-MS/MS

Although several platforms have been developed for lipidomic profiling by using LC-MS/MS, most lipid species are detected without group or class separation. Recently, a method to simultaneously quantify lipid species was established by using SFC-MS/MS [40]. We therefore analyzed the NIST plasma samples by using SFC-MS/MS to evaluate the quantified values of the lipid species detected by FIA-MS/MS using Kit-Met 2.

The average concentrations of the lipid species detected by SFC-MS/MS in the NIST plasma samples are listed in Appendix A. Separation of a class of lipid species was clearly observed from the multiple reaction monitoring (MRM) chromatograms obtained by SFC-MS/MS analysis (Appendix A). We then selected lipid species that were detected by both SFC-MS/MS and FIA-MS/MS. Most of the ratios of the concentrations of lipid species such as LPCs, PCs, and Cers between FIA-MS/MS and SFC-MS/MS were demonstrated to be highly correlated with those technologies, whereas the concentrations of SMs, hexosylceramides (HexCers), DGs, TGs, and CEs detected by the two technologies had larger differences (Figure 4). For instance, LPCs including long acyl chains exhibited larger variations in the plots caused by lower concentrations with higher CVs (%). In particular, most of the CEs except CE (16:0) were 20 times lower when detected by FIA-MS/MS than by SFC-MS/MS. The differences between the values quantified by the two methodologies might depend on lipid class separation.

#### 2.2.2. Correlation Analysis of the Quantified Values of the Lipid Species in the NIST Plasma Samples by UHPLC-FTMS and FIA-MS/MS

High-resolution MS has been widely utilized to establish determination methods for lipidomic profiling by UHPLC-FTMS [41]. Quantification is difficult due to the less than optimal stable isotope labeling of the IS, which would prevent the effects of ion suppression without chromatographic separation. Recently, a mixture of 69 stable isotope-labeled lipid species was provided to improve lipid species quantification. We therefore used the IS to quantify the lipid species in the NIST plasma samples by UHPLC-FTMS, which was performed with modifications to the previous lipidomic profiling method [42].

A total of 148 lipid species, including LPCs, PCs, Cers, SMs, HexCers, DGs, TGs, and CEs, could be annotated and quantified by UHPLC-FTMS. The annotated information and mass chromatograms of each group are shown in Appendix A and Appendix A, respectively.

The variation in the lipid species detected with FIA-MS/MS and UHPLC-FTMS similarly corresponded with the results obtained by comparing with SFC-MS/MS (Figure 5a–h). Interestingly, the values of CEs quantified with UHPLC-FTMS were higher than the ratio obtained by FIA-MS/MS but lower than those obtained from SFC-MS/MS (Figure 5h), and the values of TGs determined by UHPLC-FTMS showed a higher correlation with the values determined from FIA-MS/MS than with the results determined from SFC-MS/MS (Figure 5g).

Finally, we compared the quantified values from the three platforms with the consensus values obtained from a previous publication [33].

#### 2.2.3. Variation in the Quantified Values of the Lipid Species in the NIST Plasma Samples by FIA-MS/MS, SFC-MS/MS, and UHPLC-FTMS

A summary of the quantified values of the metabolites obtained from FIA-MS/MS, SFC-MS/MS, and UHPLC-FTMS and the consensus values based on a previous publication of NIST plasma samples is shown in Figure 6. Most LPCs and PCs could be quantified within a similar range of concentrations and corresponded with the consensus values of the previously determined results [33] (Figure 6a,b). Although values of the lipid species in each group (such as Cers, SMs, HexCers, and TGs) determined by FIA-MS/MS might be relative, the absolute quantified values have large variation among the three methodologies (Figure 6c–e,g). In contrast, the quantified values of DGs determined with the three methodologies overlapped because some lipid species overlapped and were identified without separation (Figure 6g). Notably, the values of CEs quantified from FIA-MS/MS showed large differences compared with those of the other three platforms (Figure 6h). The results indicate that it is necessary to normalize the quantified values of these lipid species with a reference material, such as the NIST and gQC plasma samples, for the analysis of a large-scale cohort. Additionally, the values of DGs analyzed by FIA-MS/MS did not show any correlation with the values from the other three platforms or references and should be used carefully as reference values.

### 2.3. Effect of Normalization with the gQC Samples for Metabolic Profiling in a Large-Scale Cohort

We assessed the normalization procedure using the values of the metabolites from the prospective cohort containing 2541 plasma samples and 33 NIST and 132 gQC plasma samples for inter- and intraplate variation within the 33-plate analysis. The plasma metabolite values quantified by Kit-Met 2 were examined by PCA to visualize plate-to-plate variations. We colored each plate and selected PC5 and PC6 to observe the time-dependent batch effects on the score plot determined by PCA, and drift could be observed among the 33 plates (Figure 7a).

We therefore introduced a normalization process to correct for interplate variation using four intermittent gQC samples, which was efficient to correct plate-to-plate drift by following the standard process using Kit-Met. Finally, the interplate variations among the 33 plates were significantly reduced on the score plots after the normalization process (Figure 7b), and the visualized variation among the cohort plasma, 33 NIST (yellow dots), and 132 gQC (orange dots) samples clearly disappeared (Figure 7a,b). We also examined the normalization process using Kit-Met 1 for the analysis of 6159 cohort plasma samples and 80 NIST and 320 gQC plasma samples. However, interplate variation in the PCA score plot could not be improved after normalization (data not shown). The average quantified values of 2541 cohort plasma samples after normalization are listed in Appendix A.

We surmised that our gQC normalization process could be a way to create a more accurate metabolic profiling database of the obtained large-scale cohort for Kit-Met analysis.

## 3. Discussion

Metabolic profiling in plasma samples has been widely performed via Kit-Met in several large-scale cohort studies, such as EPIC (European Prospective Investigation into Cancer and Nutrition) [30,43,44] and KORA (Cooperative Health Research in the Region of Augsburg) [31], and biomarker candidates have been reported to predict disease expression and progression when a previous version of this kit (AbsoluteIDQ p180 Kit, kit180) was used. In fact, kit180 stably quantified 110 metabolites in 10 μL plasma samples within a large-scale analysis in our previous study [38]. The current version, Kit-Met, increased the number of quantified metabolites by more than 600, and therefore, Kit-Met was utilized by the UHPLC-MS/MS system. We also analyzed approximately 2300 plasma samples with Kit-Met 1 in our cohort, and the quantified values were published in 2020 [6].

In the present study, we examined 6159 plasma samples to expand the reference values from 2020 to 2021 by means of a Kit-Met 1. However, the large variation in quantified values in the NIST plasma samples could be observed from the metabolites detected using Kit-Met 1. Notably, the interplate variation of a single plot of typical molecules (such as glutamine and PCaa 34:2) in NIST plasma among 80 plates detected by Kit-Met 1 was larger than that detected in 33 plates by Kit-Met 2 (Figure 2). Although the lower sensitivity of the MS system contributed to the CV differences, especially the lower concentrations of lipid species (such as Cers, HexCers, DGs, and TGs) detected using Kit-Met 1 and Kit-Met 2, interplate variation should be considered for large-scale analyses. We next established Kit-Met 2 in our large-scale cohort study to improve and continue the analysis for metabolic profiling. The CV of the quantified values clearly improved during the analysis of the 2541 plasma samples (Figure 2). We surmise that the metabolic profiles from the 2541 cohort plasma samples have the potential to be utilized to create a database including values quantified by Kit-Met 2.

Although Kit-Met, which was used in both UHPLC-MS/MS mode and FIA-MS/MS mode, has been established with an SOP, the absolute quantified values of the metabolites need to be considered. We previously demonstrated that chromatographic separation was essential to obtain the precise and accurate determination of endogenous molecules [45,46], and the separation of MRM chromatograms in UHPLC-MS/MS mode would be sufficient for the necessary level of determination (Appendix Aa). In fact, our previous 2300 cohort plasma analyses performed using Kit-Met 1 demonstrated that the quantified values of the metabolites, such as amino acids and their related compounds, in UHPLC-MS/MS mode were highly correlated with their values determined by NMR [6], which allowed reliable and accurate absolute quantified values to be obtained from biological samples. In fact, our present study demonstrated that approximately half of the 31 codetected metabolites were highly correlated with the quantified NMR values within lower CVs. However, because of the lower stability of derivatization, some metabolites (such as Cys, Arg, Lys, Lac, Orn, p-Cresol-SO_4_, HipAcid, Ind-SO_4_, and TMAO) have large variations compared with those determined using NMR. The stability and efficacy of derivatization with PITC should be considered to study the differences in the quantified values of these metabolites.

Shotgun lipidomic profiling approaches, such as direct infusion and FIA, are utilized for the rapid detection or determination of lipid species [47,48]. Kit-Mets have therefore been widely used for lipidomic profiling, and lipid species could be potential biomarkers of disease prediction. For instance, a lower abundance of SMs in plasma contributes to a high risk of MCI [49], and a lower abundance of PCs and a higher abundance of TGs are typically observed in subjects with a high BMI [30]. However, most annotated lipid species in the kit include several isobaric and isomeric compounds that cannot be separated by the applied FIA approach [50]. Indeed, the MRM chromatograms in FIA-MS/MS mode showed that these isomers were detected together, and low-abundance species were detected due to ion suppression (Appendix Ab).

In contrast, LC/MS techniques based on separation using columns, such as octadecyl silica (ODS) and hydrophilic interaction chromatography (HILIC), have been widely used for lipidomic profiling in recent studies [41,42,51]. A wide range of lipid species and isomers can be detected after chromatographic separation using an ODS column. In fact, we previously quantified lipid species by separating *sn*-1 and *sn*-2 positional isomers of lysophospholipids and sphingolipids by LC-MS/MS [52,53]. However, it is difficult to prepare optimal ISs for all detected lipid species because matrix effects cannot eliminate background ions coeluting and overlapping with the peaks of interest on the chromatogram [41]. We therefore used a mixture of 69 ISs to improve the limitations of LC/MS-based lipidomic quantification. The level of quantified values reached the consensus values from previous reports [33]. However, many lipid species still overlapped on the chromatograms, and the selection of the optimal ISs to obtain accurately quantified values was unclear. Therefore, lipid class separation is still necessary for accurate quantification of lipid species globally. Previously, HILIC-based separation was utilized to overcome this limitation [54,55]. However, the sensitivity and reproducibility of the retention times are generally reduced compared with those of ODS-based separation techniques due to the lower efficacy of electrospray ionization, which uses a high rate of water during ionization, and the lower robustness of the HILIC column [54,55].

Recently, lipid class-based separation by the SFC technique has been utilized for lipidomic profiling [56]. This methodological improvement in the quantification of individual lipid species that coelute with ISs in endogenous species conducted with MRM seems to correspond more accurately with the exact values of the lipid species [40,57]. Based on this background, the quantified values of the lipid species in FIA-MS/MS mode need to be considered and must be evaluated by other technologies, and it is essential to show the utilization of quantified values as a healthy control reference for biomarker discovery studies and precision medicine. We therefore analyzed NIST plasma samples by SFC-MS/MS to evaluate the quantified lipid species values detected by FIA-MS/MS.

The values of LPCs, PCs, Cers, DGs, and TGs quantified by SFC-MS/MS showed better correspondence with the reported consensus values than the values determined by FIA-MS/MS. However, some lipid species were detected without separation and presented in higher abundance, such as CE 18:2, CE 20:4, and CE 18:1, which might be strongly suppressed by ionization using FIA-MS/MS [33,41]. In addition, the one-point quantification method has one disadvantage of not estimating the saturation of ionization to obtain absolute values of quantification. To avoid these phenomena, tenfold diluted samples were analyzed immediately after the first analysis by SFC-MS/MS to quantify CE 18:2, CE 20:4, SM 34:1, SM 34:2, SM 36:1, SM 38:1, SM 40:1, SM 40:2, SM 42:1, SM 42:2, LPC 16:0, LPC 18:0, LPC 18:1, and LPC 18:2, which showed saturation at the first injection. Interestingly, the value of CE 20:4 was three times higher than the consensus value from a previous publication [33]. We surmise that the consensus values of the lipid species in an interlaboratory study could show a large variation between several methodologies for analysis of the same samples because the values might not consider the capacity of ionization difference between different MS systems, and the quantified values could be estimated to be lower than the absolute values of these lipid species in the NIST plasma samples. In addition, we did not evaluate the rate of false positive identifications or the correctness in applied annotations by FIA-MS/MS. Therefore, we believe that kit analysis did not produce accurate quantified values of these lipid species, such as CEs, although the results of SFC-MS/MS analysis were more accurate.

On the other hand, the LPCs and TG values quantified by FIA-MS/MS correlated with the values from other methodologies but were 1.5 times higher than those from the SFC-MS/MS and UHPLC-FTMS analyses (Figure 4 and Figure 5). For the lipid species that were potentially isomers, SFC or UHPLC methodologies using a column allow the separation of the positional isomers *sn*-1 and *sn*-2 and LPCs and TGs, as their values were estimated to be lower than the values determined by FIA-MS/MS. This result is due to interference from background ions that were derived from other lipid species and detected at the same MRM transitions without separation by FIA-MS/MS [52]. Therefore, system differences cannot be avoided by MS-based lipidomic profiling, and it is necessary to normalize the lipid species according to the values of reference materials, such as the NIST and gQC plasma samples.

Stocks et al. demonstrated the impact of the batch normalization method with the quantified values of metabolites in NIST plasma samples in both UHPLC-MS/MS mode and FIA-MS/MS mode in an interlaboratory study [13]. We therefore normalized the data from the 2541 plasma samples in the metabolic profiling study obtained by Kit-Met 2 with the quantified values of each metabolite in the gQC plasma samples by MetIDQ Oxygen software. Notably, the interplate variations were reduced after normalization in our present study (Figure 7). Although the normalization process works for several thousand assays, it is essential for evaluating the detailed variation of interplate differences. In fact, we examined the normalization process via Kit-Met 1 for 6159 cohort plasma samples to combine with the 2541 cohort plasma samples detected by Kit-Met 2, and the CV values of metabolites, especially lipid species, were improved by Kit-Met 1. For instance, the 314 lipid species in the 33 NIST plasma samples were inside 30% CV after normalization and approximately three times higher than before normalization. However, interplate variation was still observed, even after normalization by Kit-Met 1, which was shown as PCaa 34:2 and could not be improved by the visualized distribution of 6159 plasma sample analyses on the score plot of PCA. We therefore tried to create criteria to exclude the sample/s and metabolite/s in data processing and would like to present the technique using phenotypic analyses of cohort information in future studies.

We therefore surmised that our gQC normalization process could be a way to adjust the values obtained by Kit-Met analysis to obtain true representations with the minimum requirement of the criteria to exclude samples/plates and obtain the highly reliable quantified values of metabolites in large-scale sample analysis.

Bowden et al suggested that method standardization for lipidomic profiling using several platforms might be the greatest challenge, and the values of the lipid species within the same reference material obtained by different laboratories might not correspond [33]. Based on this background, the Lipidomics Standards Initiative (LSI; https://lipidomics-standards-initiative.org/, accessed on 1 August 2021) was launched in spring 2018 to propose the introduction of guidelines and standards for lipidomics; it aims to improve the overall understanding of analytical chemistry (mass spectrometric analysis) and lipid biology and should be particularly useful to researchers new to the lipidomics field [58]. The accurate values of lipid species quantified by standardized methods will be confirmed in future studies of the consortium.

However, we should deeply consider how accurate the current quantitative estimates are relative to absolute values by means of different methodologies, such as SFC-MS/MS and UHPLC-FTMS using a mixture of 69 ISs, and improve the accurate values by the gQC normalization procedure in future studies. Moreover, the reference values should be evaluated by comparison with real clinical studies to determine whether they are more significant for biological variation than previous studies [59]. In addition, even if the methodologies are better for obtaining accurate values, it is essential to evaluate the utilization of metabolic profiles for biomarker research with large-scale data, such as genome-wide association studies and multiomics analyses [60].

There are several limitations of the present study. Since 2541 cohort plasma samples were not sufficient to create a reference value database for metabolic profiling, we continued using Kit-Met. Evidently, the number of analyses will be expanded in our future studies. However, even when the same system was used, the algorithm used to calculate the quantified values was essential for the reproducibility of the assays. Indeed, the CV values were completely different because the previous version of Kit-Met 1 used the peak height of each lipid species for quantification, whereas the current version of Kit-Met 2 used the peak area in FIA-MS/MS mode for quantification. Moreover, the present study has shown a comparison of quantified results using two systems. However, these experiments in our cohort study included only Kit-Met analysis, and we did not imply quality differences in the MS systems. Interlaboratory studies, such as ring trials, are needed in the future.

## 4. Materials and Methods

### 4.1. Reagents

Acetonitrile, chloroform, isopropanol, and methanol were purchased from Kanto Chemical Co., Inc. (Tokyo, Japan) for the LC/MS analyses. Formic acid, ammonium formate (1 mol/L) and phenyl isothiocyanate (PITC) were purchased from FUJIFILM Wako Pure Chemical Corporation (Osaka, Japan). Purified water was obtained from a Milli-Q Gradient system (Millipore, Billerica, MA, USA). Ethanol was purchased from Nacalai Tesque (Kyoto, Japan). The mixture of ISs (UltimateSPLASH^TM^ ONE) and the chemical standards of lipid species were obtained from Avanti Polar Lipids Inc. (Alabaster, AL, USA) and Millipore-Sigma Ltd. (Burlington, MA, USA). Pooled normal human plasma, Na EDTA (IPLA-N, Lot: 26393), was purchased from Innovative Research, Inc. (Novi, MI, USA). The reference material (NIST SRM 1950 plasma, NIST QC) [61] was purchased from Merck Sigma-Aldrich (Darmstadt, Germany).

### 4.2. Study Population and Plasma Collection of Metabolic Profiling

For metabolome analyses, we selected a total of 8700 plasma samples, which comprised 6159 and 2541 samples for the Kit-Met 1 and Kit-Met 2 systems, respectively. All of the plasma samples were from adult participants joining the TMM project of population-based prospective cohort studies, which has more than 150,000 participants and includes the Community-based Cohort Study and Birth and Three-Generation Cohort Study in Japan. The inclusion criteria for the TMM Community-based Cohort Study were as follows. (1) For the specific health checkup site-based survey, persons were aged 40 to 74 years. (2) For the Community Support Center or the Satellite-based survey, persons were aged 20 years or more at the time of enrollment, and for the TMM Birth and Three-generation Cohort Study, pregnant women and their fetuses and children, fathers, grandparents, and other family members were included. More detailed information on the participants was described in a previous publication [35]. The cohort study and omics study were approved by the ethics committee of Tohoku University. All adult participants signed an agreement based on informed consent.

Blood samples were collected from participants with overnight fasting or morning fasting using tubes containing ethylenediaminetetraacetic acid (EDTA)-2Na. After collection, sample tubes were immediately inverted 10 times and stored at 4 °C. These sample tubes were transported to the laboratory using refrigerated containers with temperature data loggers. The transported tubes were centrifuged at 2330 × *g* for 10 min at 4 °C. The plasma fraction was transferred to a liquid handling machine and dispensed into 1.0-mL 2D barcoded screw tubes. The number of dispensed tubes was four per blood sample, the plasma volume in each tube was approximately 700 μL, and the samples were stored at −80 °C in a TMM biobank. For metabolomics analyses, the plasma sample in each dispensed tube (700 μL) was further divided into six tubes (approximately 120 μL per tube) and stored at −80 °C [37].

### 4.3. UHPLC-MS/MS Analysis

#### 4.3.1. Sample Preparation

Plasma (77 samples), gQC (4 samples), and NIST plasma (1 sample) were set out in a 96-well format according to a predefined plate layout and were prepared using the MxP^®^ Quant 500 kit (Biocrates Life Sciences AG, Innsbruck, Austria). Then, the blank calibration standard, Biocrates QC, gQC, NIST QC or sample (10 μL), was applied to each plate well. All of the sample preparation procedures followed the kit protocols, and the detailed UHPLC-MS/MS methods and conditions were previously described [6].

#### 4.3.2. Data Acquisition and Data Processing

The UHPLC system consisted of dual pumps (ACQUITY UPLC H-Class, Waters, Wilmslow, Manchester, UK) and an autosampler with a column compartment (ACQUITY UPLC I-Class). Triple quadrupole tandem mass spectrometry (MS/MS) was performed with the Xevo^®^ TQ-S system (Waters) and Kit-Met 1 for analysis of the 6159 plasma samples and with the Xevo^®^ TQ-XS system (Waters) and Kit-Met 2 for analysis of the 2541 plasma samples. The optimal UHPLC-MS/MS and FIA-MS/MS modes with all ionization parameters, ion transfer voltages/temperatures, and the detection of *m/z* pairs of precursor and product ions in MRM mode by MS were automatically set using the method in the kit. The data were collected by MassLynks 4.2 software (Waters). Quantified values (μmol/L) were calculated and normalized by their gQC values according to the manufacturer’s protocol using MetIDQ Oxygen software. The data sheet of an Excel file version 2018 was exported from the software and highlighted in 4 colors according to the kit criteria: purple: <not detected; yellow: IS out of range; light blue: <lower limit of quantification or >upper limit of quantification; and green: quantified. We selected the qualified metabolites colored green and light blue.

### 4.4. NMR Analysis

#### 4.4.1. Sample Preparation, Data Acquisition, and Data Processing

Metabolites were extracted from plasma (200 μL) by 800 μL of methanol, and the sample was suspended in sodium phosphate buffer (200 μL, 100 mM, pH 7.4) in 100% D2O containing d6-DSS (200 μM). All NMR experiments were performed at 298 K (25 °C) on a Bruker Avance III HD 600 MHz spectrometer equipped with a CryoProbe and SampleJet changer (Bruker BioSpin, Germany). Standard 1D-NOESY and CPMG (Carr-Purcell-Meiboom-Gill) spectra were obtained for each sample. All spectra were acquired with 32 scans and 32 k complex data points. All data were processed using the Chenomx NMR Suite 8.4 Processor module (Chenomx).

#### 4.4.2. Manual Quantification of the Metabolites in Plasma

Metabolites in plasma were identified and quantified using the target profiling approach implemented in the Chenomx NMR Suite 8.4 Profiler module (Chenomx). Standard 1D-NOESY spectra were analyzed for the identification and quantification of metabolites. 1D-CPMG spectra were also used to eliminate the influence of residual proteins on quantification. A typical NMR spectrum with identification of metabolites is shown in Appendix A.

### 4.5. SFC-MS/MS Analysis

#### 4.5.1. Sample Preparation

Five NIST plasma samples were prepared for extracting lipids using the Bligh and Dyer method [62]. Lipids were extracted from NIST plasma (50 μL) with 930 μL of methanol, IS-A (10 μL, Mouse SPLASH Lipidomix Mass Spec Standard, Avanti Polar Lipids Inc., Alabaster, AL, USA), and IS-B (10 μL, Avanti Polar Lipids Inc.) containing 0.050 nmol Cer d18:1 (d_7_)–15:0 and 0.050 nmol HexCer d18:1 (d_7_)–18:1. The samples were vigorously mixed for 1 min followed by 5 min of sonication. The extracts were then centrifuged at 16,000× *g* for 5 min at 4 °C, and the resultant supernatant (400 μL) was collected. After mixing with chloroform (400 μL) and water (320 μL), the aqueous and organic layers were separated by mixing and centrifugation at 16,000× *g* and 4 °C for 5 min. The organic layer (bottom, 280 μL) obtained by phase separation was dried under a nitrogen stream and stored at −80 °C until analysis. Prior to analysis, the dried sample was reconstituted in methanol/chloroform (1/1, *v/v*, 100 μL).

#### 4.5.2. Data Acquisition and Data Processing

The analytical conditions for SFC-MS/MS analysis were performed as previously described [40]. The SFC (Nexera UC system, Shimadzu) conditions were as follows: column, ACQUITY UPC2 Torus diethylamine column (3.0 mm i.d. × 100 mm, 1.7 μm particle size, Waters); injection volume, 2 μL; column temperature, 50 °C; mobile phase A, supercritical carbon dioxide; mobile phase B (modifier) and make-up pump solvent; methanol/water (95/5, *v/v*) with 0.1% (*w/v*) ammonium acetate; flow rate of mobile phase, 1.0 mL/min; flow rate of make-up pump, 0.1 mL/min; and back pressure regulator, 10 MPa. The gradient conditions were as follows: 1% B, 0–1 min; 1–75% B, 1–24 min; 75% B, 24–26 min; and 1% B, 26–30 min. The triple quadrupole mass spectrometer (TQMS, LCMS-8060, Shimadzu) analysis conditions were as follows: polarity, positive and negative ionization; electrospray voltage, 4 kV in the positive ion mode and −3.5 kV in the negative ion mode; nebulizer gas flow rate, 3.0 L/min; drying gas flow rate, 10.0 L/min; desolvation line temperature, 250 °C; heat block temperature, 400 °C; and detector voltage, 2.16 kV. The multiple reaction monitoring (MRM) parameters per one time-period were as follows: limit on number of MRM transitions, 150; dwell time, 2 ms; pause time, 2 ms; and polarity switching time, 5 ms. Data processing was performed by LabSolution software version 5.99 SP2 (Shimadzu, Kyoto, Japan).

### 4.6. UHPLC-FTMS Analysis

#### 4.6.1. Sample Preparation

The five NIST plasma samples were prepared by the Folch method [63]. Methanol (80 μL) containing IS solution was added to NIST plasma (20 μL), and the sample was homogenized with a mixer for 30 s. Then, chloroform (80 μL) was added to the sample followed by mixing for 5 min, and water (20 μL) was added to the sample and mixed for 30 s. Following centrifugation at 1500× *g* for 10 min at 4 °C, the chloroform phase (60 μL, bottom) was transferred to a sample tube. The extraction process with chloroform (80 μL) was repeated, and a second chloroform phase (60 μL) was added to the same sample tube. The final sample (120 μL) was dried by vacuum centrifugation for 30 min at room temperature. Then, the residue was reconstituted in methanol (100 μL), mixed for 3 min and transferred to the sample vial for UHPLC-FTMS analysis.

#### 4.6.2. Data Acquisition and Data Processing

The UHPLC system consisted of binary pumps, an autosampler, and a column compartment (Vanquish UHPLC system, Thermo Fisher Scientific, San Jose, CA, USA) connected with Vipers (Thermo Fisher Scientific). The UHPLC conditions were modified from a previous method [42]. Separation was performed using a metal-free C18 column (L-column2 ODS, 2.0 mm i.d. × 100 mm, 2 µm particle size; CERI, Saitama, Japan). The mobile phases consisted of (A) acetonitrile/water/ammonium formate (1 mol/L), 60/40/1 (*v/v/v*%) containing 0.1% formic acid, and (B) acetonitrile/isopropanol, 10/90 (*v/v*%) containing 0.1% formic acid. The components were separated by gradient elution. The initial condition was 30% B at a flow rate of 0.2 mL/min, followed by a linear gradient to 100% B from 2.0 min to 20.0 min, and 100% B was maintained for 10.0 min. Then, the mobile phase was returned to the initial conditions and maintained for 5.0 min until the end of the run. The total run time was 35.0 min, and the temperature of the column compartment was 45 °C.

The FTMS system was a Q Exactive Orbitrap mass spectrometer (Thermo Fisher Scientific) equipped with a heated-ESI-II (HESI-II) source. The voltages in positive and negative ion modes were 3.5 and 2.5 kV, respectively, the heated capillary temperature was 275 °C, the sheath gas pressure was 45 psi, the auxiliary gas setting was 10 psi, and the heated vaporizer temperature was 300 °C. Both the sheath gas and auxiliary gas were nitrogen. The collision gas was argon at a pressure of 1.5 mTorr. The FTMS scan type was full MS/data dependent (dd)-MS^2^. The parameters of the full mass scan were as follows: resolution of 70,000, autogain control target under 1 × 10^6^, maximum isolation time of 100 ms, and *m/z* range of 350–1050. The parameters of the dd-MS^2^ scan were as follows: resolution of 17,500, autogain control target under 1 × 10^5^, maximum isolation time of 50 ms, loop count of 5, a number of top peaks of 10, an isolation window of *m/z* 1.5, a normalized collision energy of 30, an underfill ratio of 5.00%, and an intensity threshold under 1 × 10^5^. The UHPLC-FTMS system was controlled by Xcalibur 4.2.28.14 (Thermo Fisher Scientific), and the data were collected with this software.

The lipid species were annotated with mass accuracy and MS/MS fragmentation with the analysis of typical chemical standards according to the group of lipid species, such as CEs, LPCs, PCs, Cers, SMs, HexCers, DGs, and TGs. These annotations were used to confirm the adduct ion for each group of lipid species, obtain fragment ion mass spectra information for identifying the fatty acids of lipid species, and elucidate the retention time to estimate the detection of other species within the same group of lipids that may have a different number of carbon atoms or double bonds. The quantified values were calculated from the ratio of the concentration of each optimal IS in the UltimateSPLASH^TM^ ONE chemical standards using Xcalibur software (Thermo Fisher Scientific).

## 5. Conclusions

We evaluated the quality of the quantified values of the detected metabolites in the group of 2541 plasma samples compared with the quantified values from other platforms. We demonstrated the utilization of Kit-Met in a large-scale cohort with interplate normalization by PCA. Kit-Met has been widely utilized for not only plasma but also several kids of biological specimens, and molecules detected from other specimens, such as tissue, urine, feces, bile, or cultured cells, using the kit were different from those detected in plasma. However, some lipid species values, such as those of DGs and CEs, could show large variation in absolute values. Therefore, normalization via reference materials should be considered when creating a database and used for biomarker searching in future studies of precision medicine.

## Figures and Tables

**Figure 1 metabolites-11-00652-f001:**
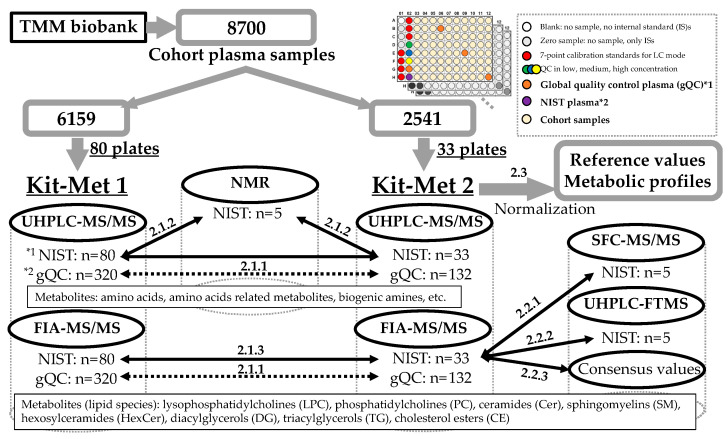
Summary of the present study. Kit-Met 1 and Kit-Met 2 were performed by the Xevo^®^ TQ-S and Xevo^®^ TQ-XS MS system (Waters, Wilmslow, Manchester, UK), respectively. Both MS systems were conducted with the UHPLC system, which consisted of dual pumps (ACQUITY UPLC H-Class, Waters) and an autosampler with a column compartment (ACQUITY UPLC I-Class, Waters). NMR, nuclear magnetic resonance; UHPLC-MS/MS, ultrahigh-performance liquid chromatography triple quadrupole tandem mass spectrometry; FIA-MS/MS, flow injection analysis MS/MS; SFC-MS/MS, supercritical fluid chromatography MS/MS; UHPLC-FTMS, UHPLC Fourier transform MS. *^1^ Metabolites in human plasma NIST^®^ SRM^®^ 1950 (Merck- Sigma-Aldrich, Parts No.: NIST1950). *^2^ Pooled normal human plasma Na EDTA (Innovative Research, Parts No. IPLA-N, Lot 26393).

**Figure 2 metabolites-11-00652-f002:**
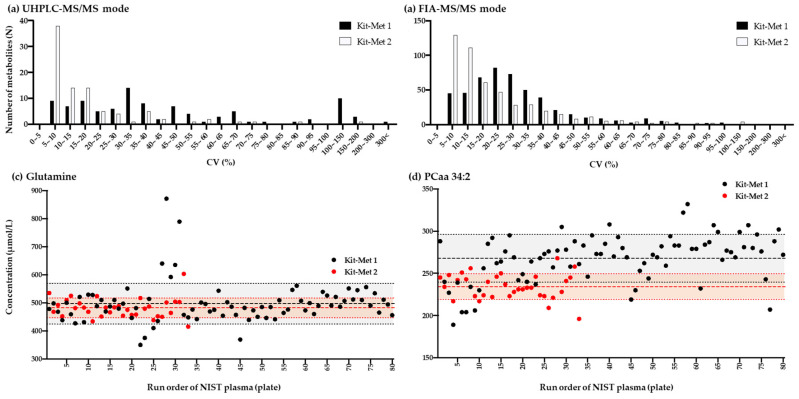
Frequency of the detected metabolites along with the range of the coefficient of variation (CV, %) in the NIST plasma samples in UHPLC-MS/MS mode (**a**) and FIA-MS/MS mode (**b**) by Kit-Met 1 (black bar) and Kit-Met 2 (white bar) and the single plots of glutamine (**c**) and PCaa 34:2 (**d**) among 80 plates evaluated by Kit-Met 1 (black dot) and 33 plates evaluated by Kit-Met 2 (red dot). The thick-dot lines are median concentration of 80 plates by Kit-Met 1 (black) and 33 plates by Kit-Met 2 (red). The filling thin-dot line areas are median concentrations ± CV.

**Figure 3 metabolites-11-00652-f003:**
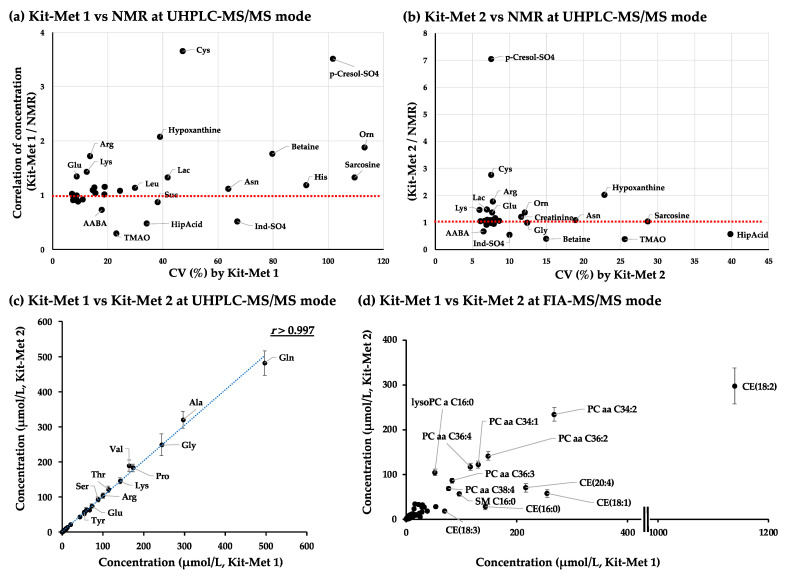
Correlation plots of metabolites detected in the NIST plasma samples by NMR, Kit-Met 1, and Kit-Met 2. (**a**) Plot of the correlation of the concentrations of 31 metabolites detected by NMR and Kit-Met 1 with CV (%) in UHPLC-MS/MS mode; (**b**) plot of the correlation of the concentrations of 31 metabolites detected by NMR and Kit-Met 2 with CV (%) in UHPLC-MS/MS mode; (**c**) plot of the concentrations (μmol/L) of 32 metabolites detected by Kit-Met 1 and Kit-Met 2 in UHPLC-MS/MS mode, where the standard deviation of each metabolite is shown for Kit-Met 2; and (**d**) plot of 273 metabolites detected by Kit-Met 1 and Kit-Met 2 in FIA-MS/MS mode. Metabolites were selected with a CV < 30% from 33 NIST plasma samples detected with Kit-Met 1 and Kit-Met 2. The dotted line shows the linear regression of each plot. The correlation of concentration was calculated by the ratio between the mean values of 80 or 33 NIST plasma samples detected by Kit-Met 1 or Kit-Met 2 and the mean values of 5 NIST plasma samples detected by NMR. The red dotted line shows a ratio of 1.0.

**Figure 4 metabolites-11-00652-f004:**
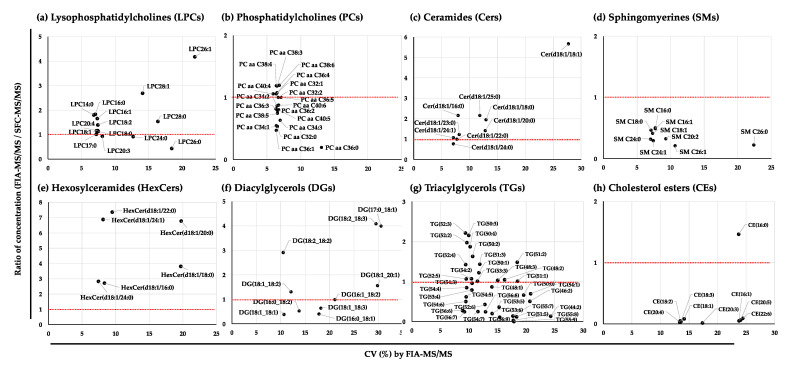
Correlation plots of lipid species in the NIST plasma samples detected by SFC-MS/MS and FIA-MS/MS. (**a**) Lysophosphatidylcholines (LPCs); (**b**) phosphatidylcholines (PCs); (**c**) ceramides (Cers); (**d**) sphingomyelins (SMs); (**e**) hexosylceramides (HexCers); (**f**) diacylglycerols (DGs); (**g**) triacylglycerols (TGs); and (**h**) cholesterol esters (CEs). Metabolites were selected with a CV < 30% from 33 NIST plasma samples detected with FIA-MS/MS. The correlation of concentrations was calculated by the ratio between the mean values of 33 NIST plasma samples detected by FIA-MS/MS and the mean values of 5 NIST plasma samples detected by SFC-MS/MS. The red dotted line shows the 1.0 correlation of the ratio.

**Figure 5 metabolites-11-00652-f005:**
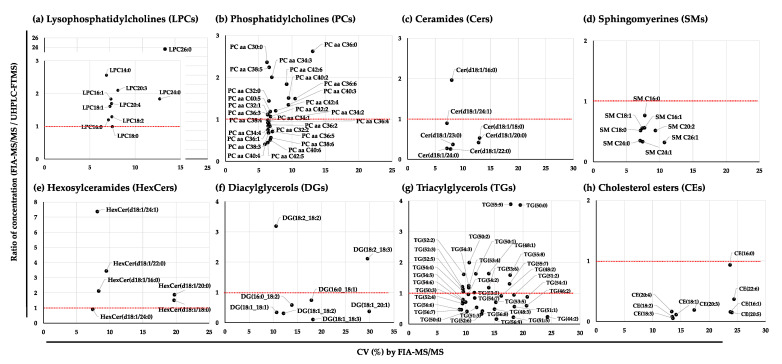
Correlation plots of lipid species in the NIST plasma samples detected by UHPLC-FTMS and Kit-Met 2. (**a**) Lysophosphatidylcholines (LPCs); (**b**) phosphatidylcholines (PCs); (**c**) ceramides (Cers); (**d**) sphingomyelins (SMs); (**e**) hexosylceramides (HexCers); (**f**) diacylglycerols (DGs); (**g**) triacylglycerols (TGs); and (**h**) cholesterol esters (CEs). Metabolites were selected with a CV < 30% from 33 NIST plasma samples detected with Kit-Met 2. The correlation of concentrations was calculated by the ratio between the mean values of 33 NIST plasma samples detected by Kit-Met 2 and the mean values of 5 NIST plasma samples detected by UHPLC-FTMS. The red dotted line shows the 1.0 correlation of the ratio.

**Figure 6 metabolites-11-00652-f006:**
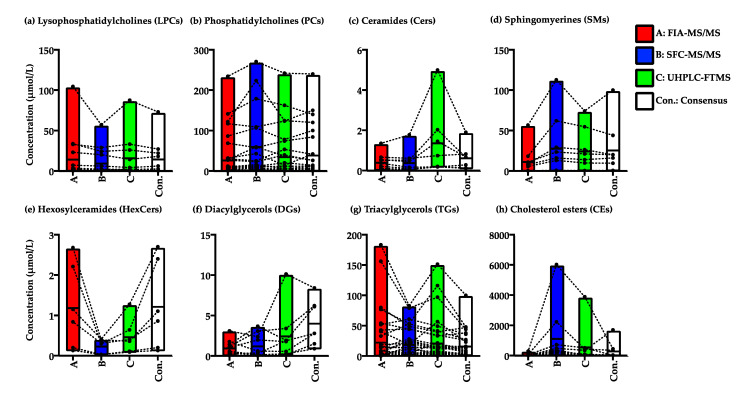
Summary of the individual values of lipid species quantified by the three methodologies and consensus values from the previous publication [33]. Methods A, B, and C show FIA-MS/MS, SFC-MS/MS, and UHPLC-FTMS, respectively. Box plot with the median intensity (horizonal bar) of lipid species separated for the lipid class, (**a**) lysophosphatidylcholines (LPCs); (**b**) phosphatidylcholines (PCs); (**c**) ceramides (Cers); (**d**) sphingomyelins (SMs); (**e**) hexosylceramides (HexCers); (**f**) diacylglycerols (DGs); (**g**) triacylglycerols (TGs); and (**h**) cholesterol esters (CEs); methods A, B, and C and the consensus method are colored in red, blue, green, and white, respectively. The black dotted lines are connected with the same lipid species between the three methods and consensus values. The consensus values were analyzed with a summary of quantitation MS platforms: triple quadrupole, quadrupole time-of-flight, and orbitrap (FTMS).

**Figure 7 metabolites-11-00652-f007:**
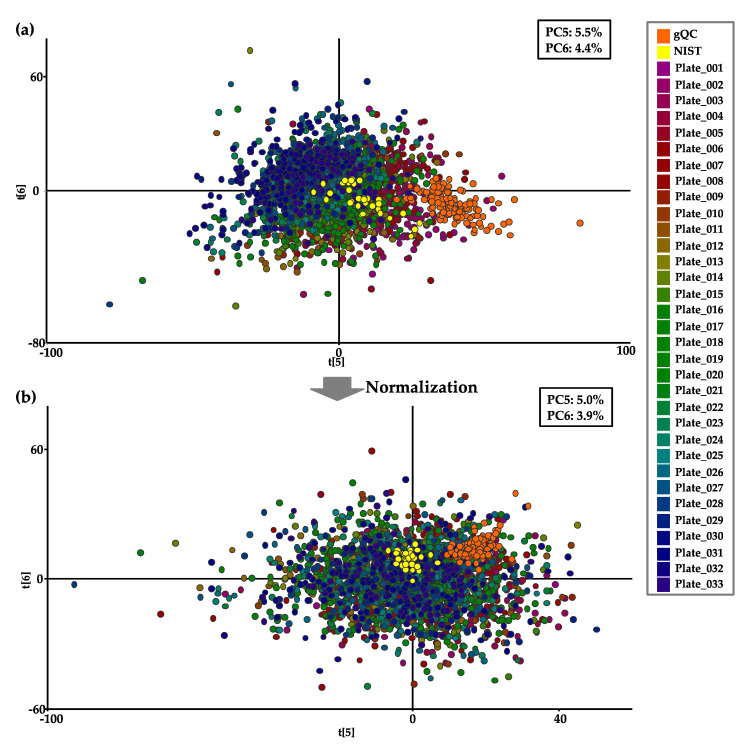
Visualization of the interplate variation in the metabolites in the cohort of 2541 plasma samples with 33 NIST plasma samples and 132 gQC plasma samples detected by Kit-Met 2 on the PCA score plot before (**a**) and after (**b**) normalization by gQC. Samples are represented by dots color-coded for the 33 continuous plates. * The NIST and gQC plasma samples are represented as yellow and black dots, respectively.

**Table 1 metabolites-11-00652-t001:** Demographic characteristics of participants. The values were mean ± SD. Averrable numbers of each parameter from database are shown in the column (N). All, all subjects; M, male; F, female; BMI, body mass index; Cre, blood test creatinine; Glc, blood test glucose.

Method		Age (N)	BMI	Cre	Glc
Kit-Met 1	All	59.6 ± 12.9 (6159)	22.7 ± 3.3 (6148)	0.690 ± 0.170 (6158)	88.4 ± 16.3 (6156)
	M	62.5 ± 12.4 (1926)	23.7 ± 2.9 (1921)	0.842 ± 0.189 (1925)	93.1 ± 19.5 (1925)
	F	58.2 ± 12.8 (4233)	22.2 ± 3.3 (4227)	0.621 ± 0.102 (4233)	86.3 ± 14.2 (4231)
Kit-Met 2	All	59.1 ± 13.8 (2541)	22.9 ± 3.3 (2539)	0.718 ± 0.289 (2541)	90.2 ± 17.5 (2539)
	M	63.7 ± 11.7 (1085)	23.7 ± 2.8 (1085)	0.849 ± 0.350 (1085)	94.7 ± 20.6 (1085)
	F	55.7 ± 14.2 (1456)	22.4 ± 3.6 (1454)	0.620 ± 0.180 (1456)	86.9 ± 14.0 (1454)

**Table 2 metabolites-11-00652-t002:** Summary of samples.

Method	Cohort Plasma Samples (77/Plate)	NIST Plasma Sample (1/Plate)	gQC Plasma Sample (4/Plate)
Kit-Met 1	6159 (80 plates *)	80	320
Kit-Met 2	2541 (33 plates)	33	132

* Seventy-nine plates were analyzed for 77 samples, one plate for 76 samples.

## Data Availability

Because of the participant consent obtained as part of the recruitment process, it is not possible to make these data publicly available. Data are available upon request, please contact the contributing author.

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
