# Peer review of "Comparison of Kit-Based Metabolomics with Other Methodologies in a Large Cohort, towards Establishing Reference Values"

_metabolites, 2021, doi:10.3390/metabo11100652_

Round 1

Reviewer 1 Report

The manuscript raises important issues which are comparison metabolomic data obtained by various analytical platforms and the need for normalization of the acquired data and proper reference values.

Due to many methodologies used in the study, the Authors should be more consistent and precise in their description. The section „Materials and Methods” should be expanded and supplemented with some key information.

Specific remarks are listed below.

Major remarks:

- Abstract – „In this study, we compared reference values from 8,700 plasma samples obtained by commercial kit-based metabolomics using ultra-high performance tandem mass spectrometry (UHPLC-MS/MS) to other methodologies.” – This is not entirely true. The Authors analyzed 8,700 plasma samples by two different UHPLC-MS/MS platforms. Results form only one platform (covering 2,541 samples) were compared with other methodologies like concluded at the end of the manuscript „We evaluated the quality of the quantified values of the detected metabolites in the group of 2,541 plasma samples compared with the quantified values from other platforms." Therefore, the sentence in the abstract is misleading. The applied workflow was also clearly described in line 116.

- More data about samples included in the study should be provided. The Authors wrote; All of the plasma samples were from adult participants joining the TMM project of population-based prospective cohort studies” (line 529). Few sentences about the project TMM and inclusion criteria should be added. Moreover, there is a lack of description of a plasma collection procedure. What type of tubes was used? Were the samples collected after overnight fasting? What about sample storage?

- More details on the methodologies used should be provided. The manuscript lacks a description of NMR experiments. 4.4. SFC-MS/MS analysis – the name of instruments and manufacturer should be provided at least.

- 4.5. UHPLC-FTMS analysis – why three NIST plasma samples were analysed, whereas five samples were analysed by SFC-MS/MS method?

Minor remarks”

- I am not convinced of the names used at work and the abbreviations introduced. The Authors used two different UHPLC-MS/MS systems but abbreviated them in a quite bizarre way as TQ-S and TQ-XS. I have doubts whether it is appropriate to shorten the trade names of instruments, that is the Xevo® TQ-S system to TQ-S and the Xevo TQ-XS system to TQ-XS. The abbreviation TQ-S says nothing about the type of method used (UHPLC-MS/MS or FIA-MS/MS).

- Additionally, the differences between the two UHPLC-MS/MS systems from Waters should be more pronounced and described in more detail in the manuscript to justify the differences in CV values between them (2.1.1.)

- Moreover, Figure 6 shows „Summary of the individual values of lipid species quantified by the 3 methodologies, including kit500 using TQ-XS (Kit500), SFC-MS/MS (SFC), and UHPLC-FTMS (UHPLC), and the reference values from reference [33].” The abbreviations suggest that TQ-XS is a completely different analytical method, whereas is a UHPLC-MS/MS system, and to be more precise, lipid profiling was performed by FIA-MS/MS method. I recommend using full abbreviations to method names (or letters A, B, C, D) instead of misleading short versions of abbreviations (it's too much of a mental shortcut). The Authors should be more consistent because SFC-MS/MS and UHPLC-FTMS are the names of analytical methods (platforms), whereas TQ-S is a trading name of an instrument. So why not use the trade names of other instruments? Is it a company (Waters) advertisement?

Moreover, the name of the method used for the determination of reference values should be also mentioned in the figure caption.

- Figure 3 - the r values should be added with each correlation analysis plot. Moreover, it is clearly demonstrated that one metabolite possessing a large concentration value is responsible for the good correlation observed (Fig. 3a, 3b, 3c). What is this metabolite?

- line 228 „We thus examined the NIST plasma samples by means of SFC-MS/MS and UHPLC-FTMS to evaluate the quality of the quantified lipid species values by kit500 using TQ-XS, and the values from the three platforms were compared with the reference values previously described.” – Reference should be added at the end of the sentence.

- line 24 „…whereas the LPC values were 1.5-fold higher” – by which platform?

- Figure 4 - the r values should be added with each correlation analysis plot.

- line 285 – „Finally, we compared the quantified values from the three platforms with the reference values based on a previous publication [33].” – What analytical method was used to determine those reference values? This information is very important in light of the results discussed and should be added to the manuscript.

- Figure 5 - the r values should be added with each correlation analysis plot.

- Figure 7 should have better quality

- line 464 – line 489 – these are results, not a discussion

- line 611 – „The black circle was mean (=0)…” – what black circle?

Reviewer 2 Report

In this paper “Addressing accuracy of reference values in a large-scale cohort by kit-based metabolomics compared to other methodologies”, the authors compared reference values from 8,700 plasma samples obtained by commercial kit-based metabolomics using UHPLC-MS/MS to other methodologies. The concentrations of metabolites were compared with the values obtained by NMR, SFC-MS/MS, and UHPLC-FTMS. The values of the amino acids were highly correlated by NMR, and lipid species, such as phosphatidylcholines and ceramides, showed a good correlation. The work authors carried out in this paper is quite extensive and has the potential to demonstrate the pros and cons of various methodologies for metabolic profiling, but data related to NMR is lacking. I recommend this work to be published after major revision in the metabolites Journal. However, I have few comments and I recommend the authors address them. My comments are below.

  1. Although several candidate biomarkers have been identified in recent studies, the reference values of metabolites have not been clarified. What does this sentence say? What disease or what is the context here? Clarify this.
  2. Did the author acquire NMR data for polar and non-polar metabolites? Please provide few spectra in the supplementary information.
  3. Separate extraction or plasma sample preparation method from UHPLC-FTMS analysis.
  4. Where is the NMR section in the methods? According to the language of paper writing, authors showed that they carried out NMR on samples but why there is no NMR analysis in the method section?

Reviewer 3 Report

This manuscript by Saigusa et al examines the robustness of a metabolomics-based kit in a large population cohort. The study examines this in the context of other assays and contrasts the performance of the technique. This is a very large study with interesting results and should be commended for its scale. The information provided is of general interest in the field. However, the way the data is presented is less than ideal and masks a lot of the general concerns in the field that should be discussed. Below are comments and suggestions from reading the manuscript.

Introduction:

  • Abbreviating the kit used in this study to KitMet 1 and 2 does make the paper a little difficult to digest, particularly with people not familiar with how these kits work and could somewhat be misleading. The authors could introduce the Q500 kit from Biocrates in the introduction and highlight that the study was run with the two different QQQ mass specs (Because of high variance in the first?), the Waters TQ-S and TQ-XS system (which I would’ve thought to be marginally different at best). In the current version, the paper initially describes Kit-1 and Kit-2 without detailing what these are. Since these are the same assays just run on an instrument with slightly higher sensitivity, one would expect similar results.
  • What was the reason for the difference in Kit-Met reported metabolites between the two instruments? Was it due to QC filtering? I believe these were both Q500 kits which are identical. Some clarification at the start might help understand the issue. Its hard to imagine this to be purely limit of detection issue when there are reports of this kit used on older instruments.
  • Some abbreviations appear unnecessary as they are not used more than once (for example, G-Met for global metabolomics is never used again). Not abbreviating these rarely used terms would make this much easier to go through. Some terms used only twice could be written in full as well, without impacting the length of the manuscript.

Results

  • The variation is astoundingly high for the FIA analysis using kit 1. While it is understandable that there would be a large effect of time (instruments getting dirty etc) for 6159 samples on variation, the Q500 kit has several QCs and checks in place and I believe have normalization techniques for plate correction. Can the authors clarify if the reported CV’s are after the standard Biocrates QC process? Were there any plates that were outliers? Any plates re-run? Some of this information seems to be missing in the manuscript. The QC process for Biocrates should be reported.
  • The authors should present a few select species (i.e. PCaa 34:2) and plot out the NIST concentration over the 6000 samples (i.e., scatter plot of concentration vs plate number), it would give an indication of the variability observed in the study in a graphical format.
  • The text in Figure 2 appears to be blurry in my edited copy.
  • Figure 3/4/5 highlights the biggest issue I have with this study. Is a little misleading in what it is trying to convey. The ‘r’ is skewed by having a metabolite of vastly difference concentrations and would always be a very high value. For example for Figure 5H, the R would be much lower if CE18:2 was excluded. What the authors should be presenting are individuals samples run between the two platforms (for example, 10 participants run on Kit-Met 1 and NMR) and compare their reported concentration and correlate them for each individual species. This would highlight how closely each instrument performs in quantifying a specific metabolite in a tested setting with various concentrations, rather than a single point.
  • The comparison and statements between Kit1 and Kit2 are not entirely fair, as there is 3-fold higher number of plates/samples run on Kit1, which tend to increase variance in large cohort mass spectrometry studies. The plot suggested for PCaa 34:2 would be a good example to highlight visually the differences in variance across the two kits.
  • Figure 6 attempts to compare the differences between 3 methods, but it is difficult to see how big the differences are as we cannot see which species is which point (aside from maybe the upper species). Plotting this figure differently is recommended (Even simple bar/dot points for each species with error bars would convey more information. In addition, plotting this on the log scale downplays the differences between the methodologies. For example, there appears to be several orders of magnitude differences between method A and method B for PC’s. This should be discussed with more detailed results presented. Is there a particular reason for this disparity?
  • Second 2.3 / Figure 7 highlights the normalization technique used. Was there a particular reason this was divided into 3 groups? The technique would probably work on all 33 plates individually. Further, this technique assumes the variance across plates are uniform for each metabolite? Have the authors tried a plate-correction by scaling each metabolite from each plate using the QC samples ? i.e. median centering. This technique is likely more appropriate in metabolomic/lipidomic studies where the variance appears different for different species.
  • Figure 7 is hard to read, more appropriate labels on the axis would help, is this PC1/PC2 and PC3/PC40 from the PCA analysis?
  • Did this technique also work for Kit1 with the QCs from the ~6000 samples? What was the CV like after this correction for Kit1 and Kit2? It seems like the authors discard the ~6000 sample lipidomic results which is somewhat unfortunate. The differences would be of interest to discuss.
  • Figure 8 is should be made larger, some of the font is quite blurry on my copy. The results are presented in a way that doesn’t highlight the power of metabolomics. Maybe a sub-figure can highlight specific species that are dramatically different between sexes/BMI? Simple bar/scatter plots can come a long way.

Discussion

  • Some new results seem to be introduced in lines 458-462, should these be in the results section?
  • The information presented in Figure 8 seems to have been glanced over. Was there any interesting findings? Seems like a waste to have such a strong population combined with metabolomics/lipidomics but not discuss any of the results. If this was focused to be a more technical paper, it would be better to have this omitted so that it’s a more focused manuscript.
  • In line 527-528, the authors highlight that the two approaches used different methods for quantification. Why is this the case? Were they not Q500 kits? Could the authors redo the quantification using area to salvage the first 6000 samples? This difference should probably be highlighted at the start as well.

Author Response

We thank the reviewer for the professional and constructive comments. Following the reviewer#3 suggestion, we have corrected several descriptions and figures, which are highlighted yellow in the revised version of the manuscript.

Round 2

Reviewer 1 Report

I am satisfied with the changes made by the Authors. I have no other comments.

Author Response

We thank the reviewer for the professional and constructive comments.

Reviewer 2 Report

Authors have responded to my comments accurately.  I have no further comments to this paper. I recommend this paper to be published in Metabolites. 

Author Response

(The authors gave the same response as above.)

Reviewer 3 Report

The authors have addressed my concerns and have markedly improved the manuscripts presentation. 

Author Response

Thank you so much.